# Gene expression drives the evolution of dominance

Christian D. Huber[1], Arun Durvasula [2], Angela M. Hancock [3] & Kirk E. Lohmueller[1,2,4]

Dominance is a fundamental concept in molecular genetics and has implications for understanding patterns of genetic variation, evolution, and complex traits. However, despite its importance, the degree of dominance in natural populations is poorly quantified. Here, we leverage multiple mating systems in natural populations of *Arabidopsis* to co-estimate the distribution of fitness effects and dominance coefficients of new amino acid changing mutations. We find that more deleterious mutations are more likely to be recessive than less deleterious mutations. Further, this pattern holds across gene categories, but varies with the connectivity and expression patterns of genes. Our work argues that dominance arises as a consequence of the functional importance of genes and their optimal expression levels.

[1] Department of Ecology and Evolutionary Biology, University of California, Los Angeles, CA 90095, USA. [2] Department of Human Genetics, David Geffen School of Medicine, University of California, Los Angeles, CA 90095, USA. [3] Department of Plant Developmental Biology, Max Planck Institute for Plant Breeding Research, 50829 Cologne, Germany. [4] Interdepartmental Program in Bioinformatics, University of California, Los Angeles, CA 90095, USA. These authors contributed equally: Christian D. Huber, Arun Durvasula. Correspondence and requests for materials should be addressed to C.D.H. (email: chuber53@ucla.edu) or to K.E.L. (email: klohmueller@ucla.edu)

The relationship between the fitness effects of heterozygous and homozygous genotypes at a locus, termed dominance, is a major factor that determines the fate of new alleles in a population, and has far reaching implications for genetic diseases and evolutionary genetics[1–4]. Several models have been theorized for the mechanism of dominance, starting with R.A. Fisher's model, which suggests that dominance arises via modifier mutations at other loci and that these loci are subject to selection[5]. In response, S. Wright argued that selection would not be strong enough to maintain these modifier mutations. He proposed a different model (termed the "metabolic theory"), later extended by Kacser and Burns, predicting most mutations in enzymes will be recessive because the overall flux through a metabolic network is fairly robust to decreasing the amount of one of the enzymes of the pathway by one-half[6,7]. Consequently, loss-of-function mutations have a more severe effect when homozygous than when heterozygous. An alternative model, posited by Haldane and further developed by Hurst and Randerson, suggested that recessivity is a consequence of selection for higher amounts of enzyme product because enzymes expressed at higher levels are able to tolerate environmental fluctuations and loss of function (LoF) mutations[8,9].

The Wright and Haldane models predict that there is a negative relationship between the dominance coefficient ($h$) and the selection coefficient ($s$), such that more deleterious mutations will tend to be recessive, while Fisher's model makes no such prediction[10]. *Drosophila* mutation accumulation lines showed evidence of this negative relationship, providing the first empirical evidence that Fisher's theory may not hold[10–12]. While the predictions of the Wright and Haldane models may be applicable to enzymes, they fail to explain the mechanism of dominance in noncatalytic gene products[13]. Further, the extent to which these estimates apply to the majority of mutations occurring in natural populations remains to be tested. While population genetic approaches to estimate the degree of dominance from segregating genetic variation exist[14,15], they have not been widely applied to empirical data.

A major challenge to studying dominance in natural populations is that $h$ is inherently confounded with the distribution of fitness effects (DFE), such that different values of $h$ and DFEs can yield similar patterns in the genetic variation data in a single outcrossing population. Here, we circumvent this challenge by developing a novel composite likelihood approach that leverages genetic variation data from outcrossing and selfing species to co-estimate $s$ and $h$. Since selection acts immediately on recessive homozygotes in self-fertilizing organisms, the genetic variation data from a selfing species allows us to discriminate between different values of $h$. Application of our approach to amino acid changing mutations in *Arabiodopsis* suggests that most mutations are recessive and that more deleterious mutations tend to be more recessive than less deleterious mutations. We then explore which mechanistic models of dominance can explain key biological properties in our data. We find that neither Fisher's model nor the metabolic theory is consistent with all of the empirical patterns we observe. Rather, our new model, which predicts that dominance can arise as the inevitable consequence of genes being expressed at their optimal levels, can match many of the salient features of the data.

variation in a population. In an outcrossing species, the main factor determining the SFS is the difference in fitness between the homozygous wild-type and the heterozygous genotype, having fitnesses 1 and $1 - hs$, respectively (Fig. 1a). This is because random mating rarely produces homozygous-derived genotypes, since deleterious mutations typically segregate at low frequencies. On the other hand, for a strongly selfing species, genotypes are predominantly in a homozygous state due to the high level of inbreeding. Thus, the main factor determining the SFS in the selfing species is the difference in fitness between the two homozygous genotypes, having fitnesses 1 and $1 - s$, respectively (Fig. 1b). Therefore, data from the outcrossing species provide information about the product of $h$ and $s$, while data from the selfing species provide information about $s$ independent of $h$. Combining information from both species therefore allows us to estimate dominance with higher accuracy than when considering either species alone. Here, we leverage this fact by developing a composite likelihood approach, which uses the SFS of the outcrossing *Arabidopsis lyrata* and the selfing *Arabidopsis thaliana* (Fig. 1c) to co-estimate the DFE and the relation between $h$ and $s$ for new nonsynonymous mutations on recently published datasets from both species (Methods)[16,17].

**Estimates of dominance**. We model the relationship between $s$ and $h$ according to Eq. (1):

$$h = f(s) = \frac{1}{\frac{1}{\theta_{\text{intercept}}} - \theta_{\text{rate}} s}, \qquad (1)$$

where $\theta_{\text{intercept}}$ defines the value of $h$ at $s = 0$ and $\theta_{\text{rate}}$ determines how quickly $h$ approaches zero with decreasing negative selection coefficient (see Fig. 1d). We chose to model the $h$–$s$ relationship in this manner to allow for more deleterious mutations to be more recessive than less deleterious mutations, as suggested by experimental data[11–13]. We then extended the Poisson random-field model of polymorphisms[14] for estimating the two parameters of this relationship, $\theta_{\text{intercept}}$ and $\theta_{\text{rate}}$ (see Methods for further details). To account for the effects of changes in population size on the nonsynonymous SFS that might confound estimates of selection, we first estimate a three-epoch demographic model using the synonymous SFS[18]. The shape and scale parameter of a gamma distributed DFE ($\Theta_{\text{DFE}}$) and the rate and intercept parameter of the $h$–$s$ relationship, $\Theta_{\text{h}} = \{\theta_{\text{intercept}}, \theta_{\text{rate}}\}$, are then estimated conditional on the estimated demographic model. We then use the Poisson likelihood to estimate the combined vector of parameters $\{\Theta_{\text{DFE}}, \Theta_{\text{h}}\}$ according to Eq. (2):

$$L(\Theta_{\text{DFE}}, \Theta_{\text{h}} | \Theta_{\text{D}}, \theta, X_i)$$
$$= \prod_{i=1}^{n-1} \frac{\text{E}[X_i | \Theta_{\text{D}}, \Theta_{\text{DFE}}, \Theta_{\text{h}}, \theta]^{X_i}}{X_i!} e^{-\text{E}[X_i | \Theta_{\text{D}}, \Theta_{\text{DFE}}, \Theta_{\text{h}}, \theta]} \qquad (2)$$

Here, $\Theta_{\text{D}}$ is a vector of demographic parameters, $X_i$ is the count of SNPs with frequency $i$ in the sample (the entries of the SFS), $\theta$ is the population mutation rate, and $n$ is the sample size. To combine data from the outcrossing species *A. lyrata* with data from the selfing species *A. thaliana*, we compute the combined log-likelihoods (LL) over both datasets by summing over the

## Results
### Inference of dominance using inbred and outbred populations.
We propose to increase power for estimating dominance by combining data from an outcrossing species with data from a selfing species. We use the distribution of allele frequencies in a sample, or site frequency spectrum (SFS), as summary of genetic

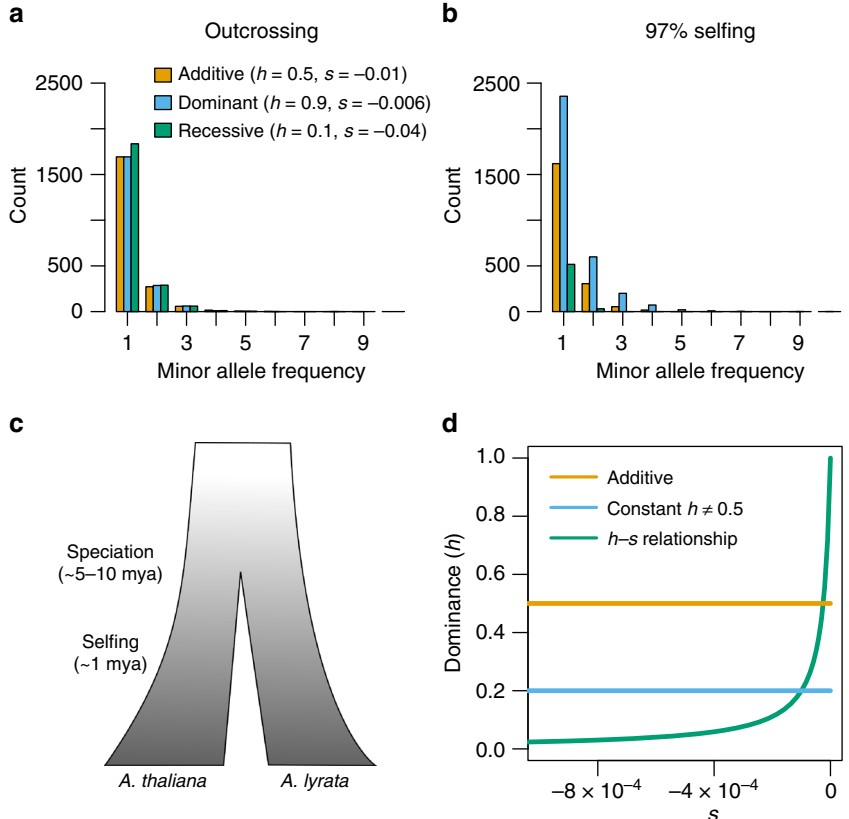

**Fig. 1** The effect of dominance and mating system on the site frequency spectrum (SFS). **a** The SFS from an outcrossing species simulated under different DFEs and $h$ values. Note that different combinations of DFEs and values of $h$ yield similar SFS. **b** The SFS for the same DFEs and values of $h$ as in **a** for a highly selfing species. Differences in $h$ result in large differences in the SFS in selfing species, allowing us to reliably co-estimate the DFE and $h$. **c** A schematic of the species history between *A. thaliana* and *A. lyrata*. **d** Examples of the relationship between $h$ and $s$ under the three different models of dominance tested here

individual log-likelihoods:

$$
\begin{aligned}
\mathrm{LL}&\left(\boldsymbol{\Theta}_{\mathrm{h}}, \boldsymbol{\Theta}_{\mathrm{DFE}} | \mathbf{SFS}_{\mathrm{O}}, \mathbf{SFS}_{\mathrm{I}}, \boldsymbol{\Theta}_{\mathrm{D,I}}, \theta_{\mathrm{I}}, \boldsymbol{\Theta}_{\mathrm{D,O}}, \theta_{\mathrm{O}}\right) \\
&= \mathrm{LL}_{O}\left(\boldsymbol{\Theta}_{\mathrm{h}}, \boldsymbol{\Theta}_{\mathrm{DFE}} | \mathbf{SFS}_{\mathrm{O}}, \boldsymbol{\Theta}_{\mathrm{D,O}}, \theta_{\mathrm{O}}\right) \qquad (3) \\
&+ \mathrm{LL}_{I}\left(\boldsymbol{\Theta}_{\mathrm{h}}, \boldsymbol{\Theta}_{\mathrm{DFE}} | \mathbf{SFS}_{\mathrm{I}}, \boldsymbol{\Theta}_{\mathrm{D,I}}, \theta_{\mathrm{I}}\right)
\end{aligned}
$$

Finally, we infer the maximum likelihood parameter values for three different dominance models (i.e. additive model, constant $h$ model, and $h$–$s$ relationship model; see Fig. 1d), and compute the likelihood ratio test statistic ($\Lambda$) to compare between the models (see Methods for further details).

We find that a model where mutations are slightly recessive (inferred $h = 0.46$) results in a significantly better fit than assuming a model where all mutations are additive (Fig. 2a). The third model allows $h$ to depend on $s$ (Fig. 1d), and we infer that this model fits the SFS significantly better than a model with a constant $h$ ($P < 1 \times 10^{-15}$; Fig. 2a; see Methods). This model is broadly consistent with previous experimental studies in flies and yeast[10,13], as well as with a QTL mapping study on the contribution of alleles at evolutionary constrained sites to variation in fitness-related traits in inbred lines of maize[19]. Importantly, mutations that are more deleterious also tend to be more recessive (Fig. 2b). For example, we find that mutations with $s < -0.001$ have an $h < 0.025$, suggesting that even moderately deleterious mutations are quite recessive. Lastly, we tested the effect of assuming alternative functions for the $h$–$s$

relationship. In particular, we tested (1) a logistic function, and (2) the effect of constraining $\theta_{\text{intercept}}$ to a value of 0.5, i.e. assuming that almost-neutral mutations are additive. Although assuming different functional relationships between $h$ and $s$ introduces some uncertainty in the inferred dominance coefficient for mutations with selection coefficients between $-5 \times 10^{-4}$ and 0, the inference of the dominance coefficient for deleterious mutations with $s < -5 \times 10^{-4}$ is robust to the assumed functional form (Fig. 3a). Notably, irrespective of the functional form, we observe strong statistical support for a negative relationship between $h$ and $s$ (Supplementary Table 3). Further, this result is robust to assuming a DFE that allows for a proportion of neutral mutations (DFE and neutral in Fig. 3a). However, because very strongly deleterious mutations ($s < -0.01$) are unlikely to be segregating in the data, we have limited resolution to infer the dominance effects for such mutations. We also tested a model where $h$ converges to one instead of zero as mutations become more deleterious, but found that this model fits the data significantly worse (Supplementary Note 1).

**Robustness of inference**. To determine whether our statistical framework is sensitive to certain confounders and can reliably distinguish between competing models, we carried out extensive forward simulations based on the demographic models inferred from our data (see Methods; Supplementary Table 1, Supplementary Fig. 1). The distribution of the likelihood ratio test (LRT) statistic in simulations where all mutations were additive resembled the predicted asymptotic chi-square distribution when comparing the constant $h \neq 0.5$ model to the additive model

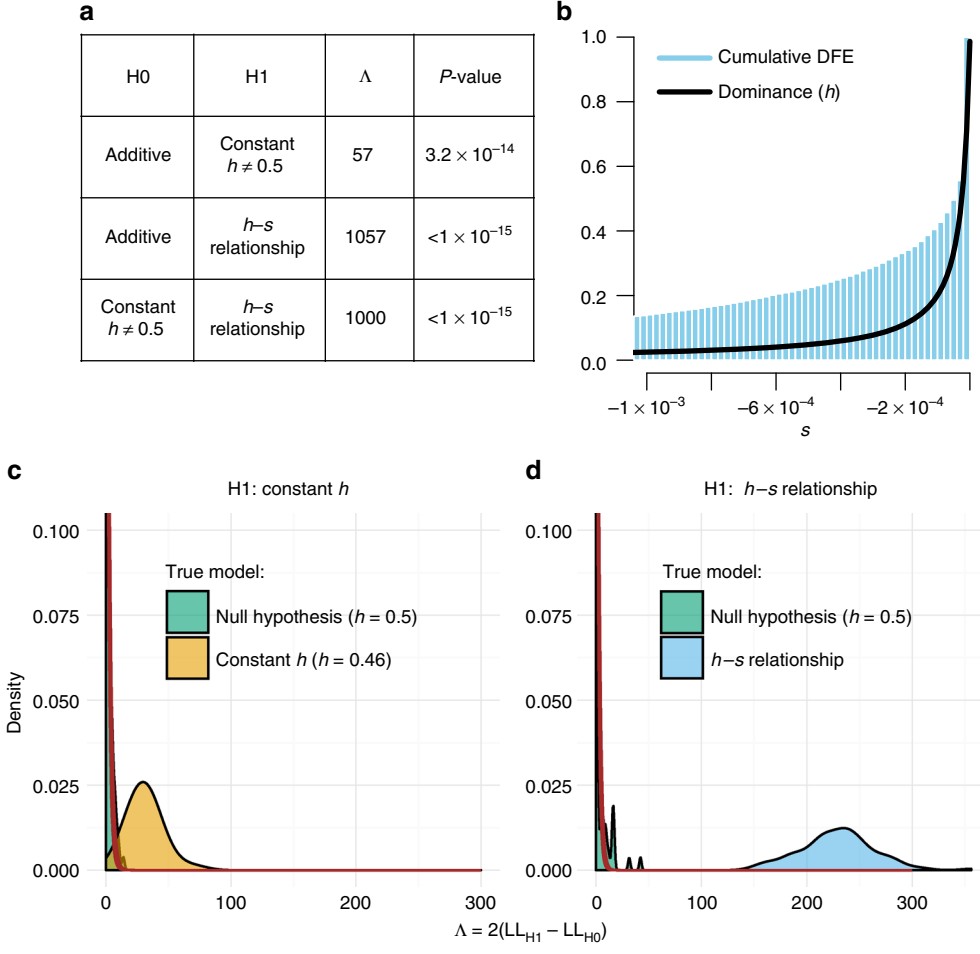

**Fig. 2** Genome-wide estimates of dominance. **a** Likelihood ratio test statistics ($\Lambda$) and $P$-values when comparing different models of dominance. The $h$–$s$ relationship fits the data significantly better than the additive model and significantly better than a model with a single dominance coefficient. **b** Inferred relationship between $h$ and $s$ based on whole genome data. More nearly neutral mutations tend to be more dominant than strongly deleterious mutations. **c**, **d** Simulations demonstrating the performance of our inference procedure. **c** Likelihood ratio tests comparing a constant $h$ model to an additive model. When data are simulated under an additive model (green), $\Lambda$ nearly follows a chi-square (1 $df$) distribution (red line). However, when the data are simulated under a model with $h = 0.46$ (tan), the distribution of $\Lambda$ is substantially larger, indicating excellent statistical power. **d** Likelihood ratio tests comparing the $h$–$s$ relationship model to an additive model. When data are simulated under an additive model (green), $\Lambda$ nearly follows a chi-square (2 $df$) distribution (red line). However, when the data are simulated under the $h$–$s$ relationship model (blue), the distribution of $\Lambda$ is substantially larger, indicating excellent statistical power

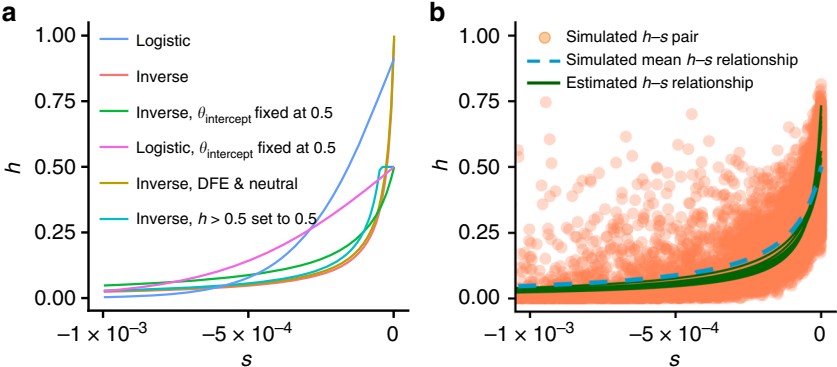

**Fig. 3** Robustness of the inferred $h$–$s$ relationship. **a** The inferred $h$–$s$ relationship depends on the assumed functional form of the relationship and on assumptions regarding the DFE. However, all estimated curves converge to strong recessivity for selection coefficients $s < -0.0005$ (see also Supplementary Table 3). The inverse $h$–$s$ relationship is defined by Eq. (1), the logistic relationship is defined by the formula $h = \theta_{\text{intercept}} \left( \frac{1 + e^{-\theta_{\text{offset}}}}{1 + e^{\theta_{\text{rate}}|s| - \theta_{\text{offset}}}} \right)$. **b** Substantial variation in the dominance coefficient $h$ for particular values of $s$ has only a modest effect on the estimation of the $h$–$s$ relationship. The estimated curves (green; 10 replicates) closely follow the simulated mean relationship between $h$ and $s$ (dashed blue line). However, the estimated intercept parameter, defining the dominance of almost-neutral mutations, is upwardly biased. Orange dots denoted individual $h$ and $s$ coefficients for mutations

(df = 1, Fig. 2c), as well as when comparing the $h$–$s$ relationship model to the additive model (df = 2, Fig. 2d). Importantly, none of the LRT statistics were as large as those seen empirically (Fig. 2a), suggesting a conservative simulation-based $P$-value < 0.01. When simulating data under the constant $h$ model ($h$ = 0.46, Fig. 2c), as well as the $h$–$s$ relationship model, we find that the distribution of the LRT statistic is strongly shifted to larger values compared to the constant $h$ = 0.5 model (Fig. 2d). These simulations suggest we have excellent power to distinguish between models given the demographic history, sample size, and amounts of genetic variation present in these species.

Our inference model assumes no variation in the dominance coefficient $h$ conditional on a given selection coefficient $s$. In reality, for any given $s$, there might be a distribution of $h$ values instead of a single value of $h$. To test how this additional variation in $h$ might affect our inference, we simulated data under a model that assumes additional variance in $h$ by sampling $h$ from a beta distribution with a fixed standard deviation of 0.1 (see Fig. 3b). The model used to simulate the data further assumes that the true $\theta_{intercept}$ is 0.5. When estimating the $h$–$s$ relationship parameters from the simulated data, we find that the estimated curves fall well within the point cloud of $h$ and $s$ values, and reflect the increase in recessivity with increasing deleteriousness of mutations fairly well. This result suggests that our estimates of the mean $h$–$s$ relationship for moderately deleterious mutations are robust to variation in $h$. However, the estimated $\theta_{intercept}$ parameter is biased to larger values than the true expected value of 0.5. Thus, our estimates of $\theta_{intercept}$ close to one in the *Arabidopsis* data might reflect a large variance in the dominance coefficient for almost-neutral mutations, even though on average those mutations might be additive. Therefore, we cannot conclude from our results that almost neutral mutations are mostly dominant.

Lastly, it is unlikely that our inference of the $h$–$s$ relationship is due to differing DFEs between *A. thaliana* and *A. lyrata* (Supplementary Note 1). First, we see significant support for an $h$–$s$ relationship over an additive or constant $h$ model even when basing our inference solely on the outcrossing *A. lyrata* data (Supplementary Fig. 2, Supplementary Table 2). Second, a purely additive model with unique DFEs in both species fits significantly worse than a model with the same DFE, but an $h$–$s$ relationship (Supplementary Table 4). In sum, it is unlikely that our conclusion of extensive recessivity of mutations and the relationship between dominance effects and selective effects is driven by artifacts of our inference procedure.

**Catalytic and structural genes**. We next sought to test which theoretical model of the basis of dominance can explain our data. Fisher's theory for the evolution of dominance predicts that $h$ should show no relationship to the degree of deleteriousness of a mutation[5,10]. Our finding of the $h$–$s$ relationship is not consistent with this theory. The metabolic theory[7] predicts that mutations in catalytic genes ought to be more recessive than those in genes unlikely to be involved in enzyme kinetics. We classified genes based on gene ontology (GO) category and inferred the DFE and $h$ for specific gene sets (Methods; Supplementary Tables 4 and 5). Overall, we find that catalytic genes display similar patterns of polymorphism (Supplementary Fig. 4) and an $h$–$s$ relationship, as seen genome-wide (Fig. 4a). Genes encoding structural proteins (herein "structural genes"), which are unlikely to be involved in enzyme kinetics, however, show a higher proportion of rare variants in the SFS (Supplementary Fig. 4) and appear to be less recessive than catalytic genes (Fig. 4a). In other words, for a given selection coefficient, mutations in catalytic genes tend to be more recessive than those in structural genes. On the surface, this

finding appears to support the prediction of the metabolic theory of dominance. However, we infer that the $h$–$s$ relationship model fits the structural genes better than the constant $h$ model or the additive model (Fig. 4c, Supplementary Table 4). Thus, even structural genes show evidence of recessive mutations, which is not predicted under the metabolic theory model. We note that this finding has previous experimental support in yeast[13,20].

**Expression level and connectivity**. To investigate other mechanisms that could lead to recessive mutations in structural genes, we classified genes based on their expression level and degree of connectivity in networks (Methods). Overall, we found that structural genes tended to be more highly expressed and have more network connections than other types of genes (Fig. 4b). We next tested whether the parameters of the $h$–$s$ relationship differed across these different functional categories (Fig. 4c, d, Supplementary Figs. 7 and 8, Supplementary Tables 4 and 5). While the $h$ intercept ($\theta_{intercept}$) did not differ across any of the categories (Fig. 4d, Supplementary Fig. 7), we found that the $h$–$s$ decay rate ($\theta_{rate}$), or slope, of the relationship between $h$ and $s$ did vary across some groupings. Specifically, the decay rate was significantly larger for catalytic genes than for any of the other categories, again indicating that mutations in these genes tend to be more recessive than those in other genes. Genes that were more highly expressed and those that tended to be more connected had a smaller decay parameter, indicating that mutations in these genes tended to be more additive (Fig. 4c, d). Strikingly, we could not reject a model where structural genes had the same decay parameter as highly connected genes, or non-structural genes that are both highly connected and have high levels of expression (Fig. 4d). These results argue that structural genes do not appear to have a unique $h$–$s$ relationship. Rather, they share the properties of other genes that are both highly connected and have a high level of expression.

**A new model for the evolution of dominance**. Our results motivate further development of a more general model for dominance. A recently developed fitness landscape model of dominance broadly predicts the recessivity of mutations as a consequence of stabilizing selection on fitness-related phenotypic traits[21]. However, this model does not explain the observed variation in dominance across different levels of gene expression. Therefore, we extended a model by Hurst and Randerson[9], where dominance is directly related to gene expression. Here, higher gene expression leads to higher fitness, but the gain from increasing gene expression is lower for higher levels of gene expression than for lower levels of gene expression (diminishing returns function). For enzymatic genes, this relationship was shown to be a consequence of metabolic pathway dynamics, assuming that the output of the system (flux) is directly related to fitness[7]. For genes encoding structural proteins, it is imaginable that after enough protein is produced to build certain structures in the cell or the extracellular matrix, additional protein does not improve its functional role any further.

To formalize such a type of diminishing returns function, Hurst and Randerson assume a simple functional relationship between expression level and fitness, $f(x) = x/(1 + x)$, where $x$ is the expression level (arbitrary units), and $f$ is the fitness. Further, they assume that per unit of $x$, there is a cost $c$ associated with gene expression. In biological systems, these costs could be related to spending cellular resources (amino acids and nucleotides), allocation of cellular machineries (RNA polymerase and ribosome), or energy consumption[22]. The expression cost $c$ is included as a parameter that quantifies the reduction in fitness per unit of gene expression, such that $f(x) = x/(1 + x)(1 - cx)$. For

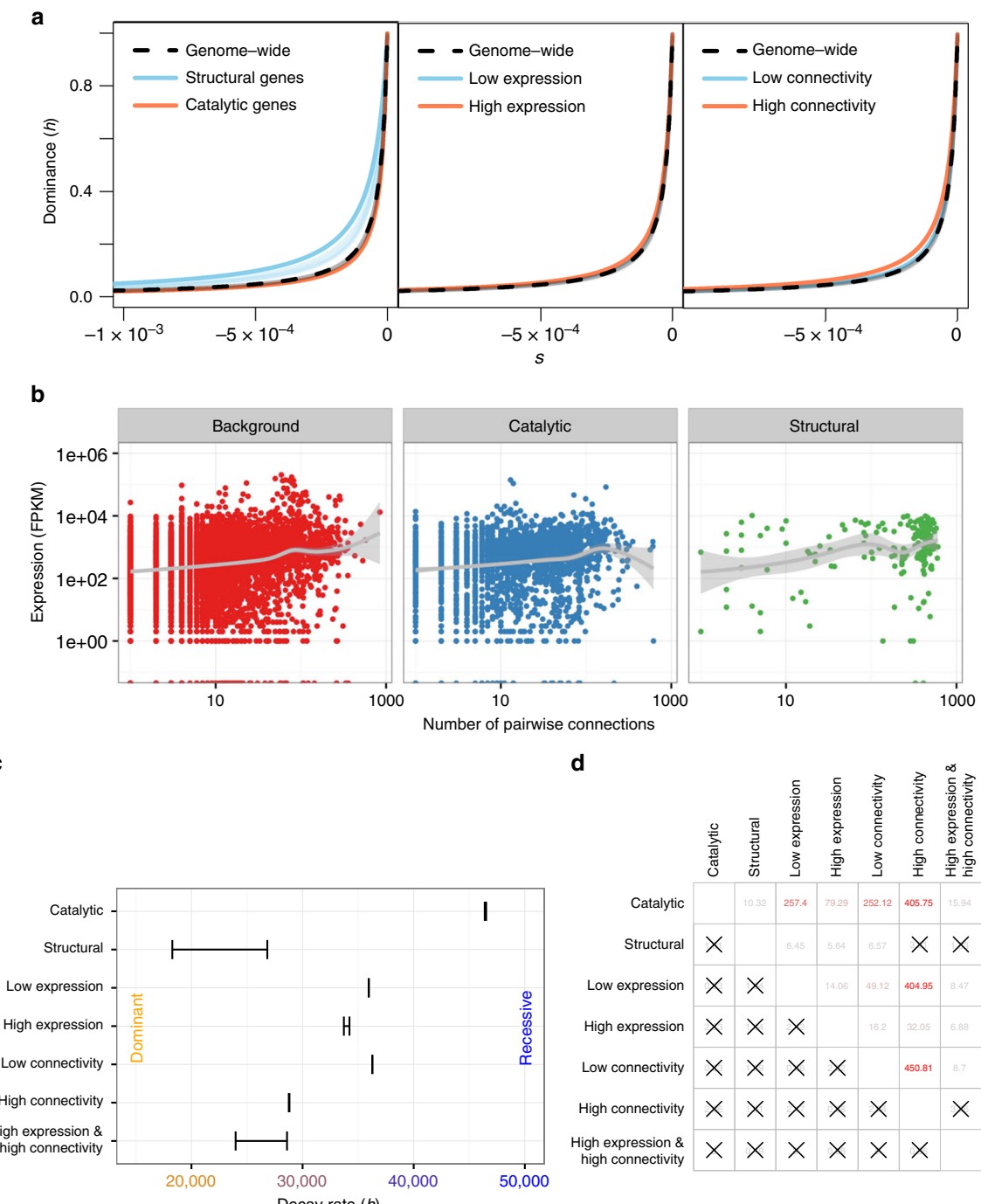

**Fig. 4** Distribution of dominance per gene category. **a** *h*–*s* relationship inferred for different gene categories. Bootstrap replicates are shown in lighter colors and in gray for the genome-wide estimates. The blue lines in the middle and right panel strongly overlap with the gray lines of the genome-wide estimates. **b** Expression profiles are correlated with gene connectivity. Note that structural genes have higher connectivity and expression than do other types of genes. Background refers to genes not in catalytic or structural GO categories. **c** Differences in the decay rate of *h* ($\theta_{rate}$) across gene categories. 95% confidence intervals (CI) are shown. Larger decay rates indicate that for a given value of *s*, mutations tend to be more recessive. **d** Z-scores for tests of differences in decay rate (upper triangle) and intercept (lower triangle) between different categories of genes. Color indicates degree of significance (red is more significant). Comparisons not significantly different after Bonferroni correction are denoted by "X"s

simplicity, we assume that *c* per unit of gene expression is the same for every gene. However, key features of the model are highly robust to different values of *c* such that variation in *c* between genes should not affect our conclusions (Supplementary Figs. 12 and 13).

We extend the model of Hurst and Randerson in two ways (see also Fig. 5a). First, the Hurst and Randerson model assumes that

the fitness at zero expression level is zero. However, experiments in bacteria, yeast, and a number of other organisms have shown that a considerable proportion of genes are non-essential, such that fitness would not reduce to zero when the gene is not expressed[23]. We include an *intercept* parameter in the model that determines the fitness when the gene is not expressed. An *intercept* close to one indicates that the gene is non-essential and

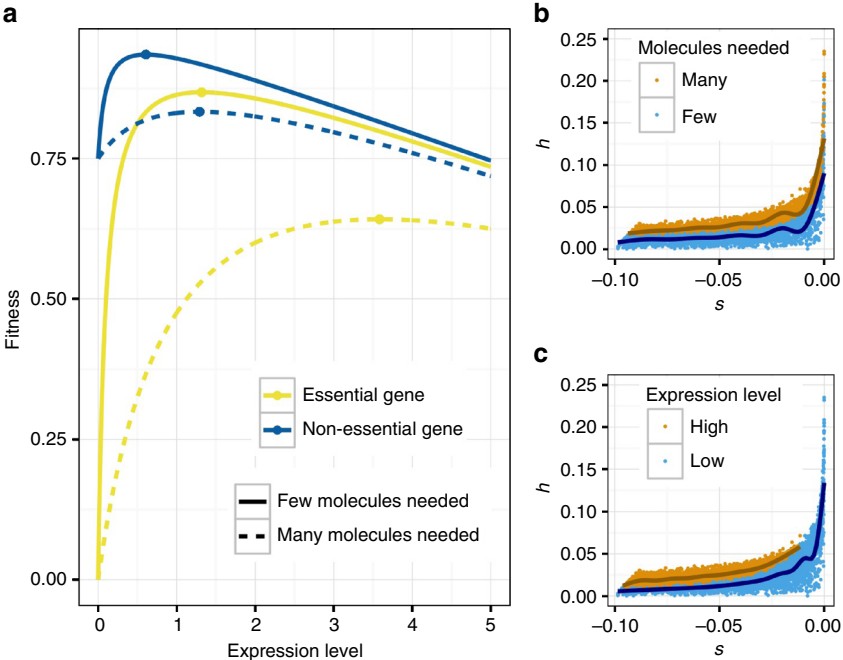

**Fig. 5** A new, comprehensive model for the evolution of dominance. **a** The relationship between fitness and expression level (arbitrary units). A fitness cost for increasing gene expression is assumed (see the main text). **b** Predicted $h$–$s$ relationship when many molecules (orange) and few molecules (blue) are needed. **c** Predicted $h$–$s$ relationship when the expression level is high (orange) and low (blue). Note that the patterns predicted in **b**, **c** mirror those seen empirically in our analysis

can be removed with only little reduction in fitness, whereas a value close to zero indicates that the gene is essential for survival or reproduction. Second, we add a *scale* parameter that allows for varying rates of increase in fitness with expression level (Supplementary Fig. 11). We define the *scale* parameter as the expression level at which fitness is exactly in the middle between the fitness at zero expression and at infinite expression (assuming no expression costs). In biological terms, this parameter is related to the amount of protein needed by the organism to function properly. For structural proteins, many molecules might be needed to build structures in or out of the cell, which would be reflected in a large *scale* parameter. For enzymatic proteins, a single protein can catalyze the same chemical reaction over and over again, thus only a small amount of molecules might be needed and the *scale* parameter would be small. The relation between expression level and fitness is then a function of $c$, *intercept*, and *scale*:

$$f(x) = \frac{(x + intercept \times scale)(1 - c \times x)}{x + scale} \quad (4)$$

The optimal gene expression under this model can be computed by setting the derivative of $f(x)$ to zero and solving for positive $x$:

$$x_{opt} = \frac{\sqrt{scale \times c \times (1 - intercept) \times (1 + scale \times c)}}{c} - scale \quad (5)$$

We assume that gene regulatory sequence is optimally evolved, such that genes are expressed at the level $x_{opt}$ (Eq. (5)). Theoretical and empirical arguments suggest that selection should be highly effective in keeping gene expression close to its optimal value[24], although we also study the model under suboptimal expression (see below). Next, we investigate the fitness effect of gene mutations that cause the protein to be

non-functional. If the mutation is heterozygous, then the amount of functional protein is only half of the amount in the wild-type homozygous genotype. If the mutation is homozygous, then no functional protein is produced. The fitness consequences of heterozygous mutations are computed by setting gene expression $x$ to $x_{opt}/2$ in Eq. (4). The fitness consequences of homozygous mutations are computed by setting $x = 0$. The selection coefficient $s$ and the dominance coefficient $h$ are then defined as

$$s = \frac{f(0) - f(x_{opt})}{f(x_{opt})} \quad (6)$$

$$h = \frac{f(x_{opt}) - f(\frac{x_{opt}}{2})}{f(x_{opt}) - f(0)} \quad (7)$$

Both $s$ and $h$ are determined by three parameters: $c$, *intercept*, and *scale*. We can investigate the relationship between $s$ and $h$ as a function of these three parameters (Supplementary Fig. 12). Suboptimal gene expression, where the expression level is at 80% of its optimal value, leads to qualitatively similar behavior as when assuming optimal gene expression (Supplementary Fig. 13).

**The model predicts key patterns in the data.** Simulations under our model (see Methods) recapitulate the key features seen in our empirical data (Fig. 5b, c). First, the simulations show a negative relationship between $h$ and $s$. More strongly deleterious mutations are more recessive than less deleterious mutations (Supplementary Fig. 12). Note that this is a consequence of selection for optimal gene expression, not because of direct selection on a dominance modifier. Direct and indirect models of selection for dominance were criticized by Orr, who has noted that a predominantly haploid organism would not be able to evolve dominance[25]. In at least one such organism, dominance of mutations

is observed, arguing against models of selection for dominance[25]. However, our model does not rely on evolution in a diploid organism, since it does not rely on selection happening only in the diploid state (see also Hurst and Randerson[9]). It is thus consistent with Orr's finding.

Second, although the model predicts that mutations are recessive, mutations become slightly less recessive when increasing the scale parameter, i.e. when increasing the optimal expression level of the gene. This predicts that mutations in genes with high optimal gene expression (many molecules are needed) would be more additive than genes with low optimal gene expression (few molecules are needed). This prediction matches our empirical analyses where gene sets with high expression level and/or high connectivity (i.e. many molecules needed), tend to be more additive compared to gene sets having low expression level and/or low connectivity (i.e. only few molecules needed) (Fig. 4c).

## Discussion

Overall, our work provides a fine-scale molecular population genetic demonstration, using genetic variation data from natural populations, that more deleterious mutations tend to be more recessive than less deleterious mutations. Further, we find that existing models of the mechanistic basis of dominance, such as Fisher's modifier model[5], the metabolic theory model[6,7], or newer models based on fitness landscapes[21] do not explain patterns of dominance that we see across all types of genes. Instead, our results support a more general model for the occurrence of dominance with testable predictions about its mechanism. Specifically, our findings suggest that dominance and the $h$–$s$ relationship arose as a natural outcome of the functional importance of genes and their optimal expression levels. In addition, under our model, dominance can evolve in haploid organisms, passing a previous test of the evolution of dominance that rejected both Fisher's and Haldane's original models[25].

Our findings have implications for evolutionary and medical genetic studies. First, many deleterious mutations tend to be recessive, and may accumulate in heterozygotes and be maintained in populations, which could increase the role of population history in affecting patterns of deleterious mutations and the genetic load[1,2]. Second, the location of a gene in a biological network and optimal expression level will influence both the selection coefficient and degree of dominance of that mutation, indicating that mutations in certain genes may be more prone to having fitness effects and being potentially involved in complex traits.

## Methods

**Data**. We collected sequencing data for 13 *A. lyrata* plants from Novikova et al.[16] and sequencing data for 16 *A. thaliana* plants from Durvasula et al.[17]. We aligned accessions to their respective genomes (*A. thaliana* to TAIR10[26] and *A. lyrata* to the JGI reference sequence v1.0[27]) using BWA-MEM (BWA 0.7.7-r441)[28] with a penalty of 15 for unpaired read pairs. We removed duplicated reads using Picard v2.7 and performed local indel realignment using Genome Analysis Toolkit (GATK v3.6) IndelRealigner[29]. We called SNPs using UnifiedGenotyper and filtered variants using the recommendations from GATK:

QualByDepth < 2.0 || FisherStrand > 60.0 || RMSMappingQuality < 40.0 || MappingQualityRankSumTest < −12.5 || ReadPosRankSum < −8.0 || StrandOddsRatio > 3.0 || HaplotypeScore > 13.0

We annotated SNPs using SnpEff v4.3a[30]. We used gene annotations (TAIR10) to filter only coding sequences (CDS) and created site frequency spectra (SFS) for synonymous and nonsynonymous variants separately. We calculated folded SFSs in order to avoid assigning an ancestral allele, which is difficult to do in these species due to extensive genome rearrangements[27]. We downsampled the SFS in *A. lyrata* from 13 entries to 11 using a hypergeometric downsampling scheme[31].

We ensured that population structure did not affect our frequency spectra by performing principal components analysis (PCA) and checking the distribution of pairwise differences between samples. We removed samples that were highly related within each species as determined by outliers in the number of pairwise

differences and individuals that cluster very closely on the PCA run on the genotypes[32] (Supplementary Fig. 3). When two accessions were closely related, we retained one individual selected at random. For the *A. thaliana* dataset, we removed samples 35601, 35513, 35600, 37469 and for the *A. lyrata* dataset, we removed samples SRR2040788, SRR2040795, and SRR2040829.

We annotated each coding site according to the gene name and GO term and subsetted the data into different GO term categories to perform our inference of dominance and the DFE separately on these categories. We annotated each gene based on connectivity and gene expression. Connectivity was determined by the STRING database v10[33]. We downloaded the *A. thaliana* (organism 3702) protein network data and restricted our analysis to high confidence (>0.7) interactions. Connectivity is then equally subdivided into three categories: low connectivity, intermediate connectivity, and high connectivity (e.g., Fig. 4). We obtained expression data for *A. thaliana* from the 1001 Epigenomes project (NCBI GEO: GSE80744;[34]), which provides a processed read count matrix for each gene across all accessions. We obtained the median expression value across all accessions, and arrived at a single value for each gene. Expression level is then equally subdivided into three categories: low expression, intermediate expression, and high expression (e.g., Fig. 4).

**Models of dominance and likelihood ratio test**. We test three different models of the relationship between the selection coefficient of a mutation ($s$) and the dominance coefficient ($h$). Here, $s$ and $h$ are defined such that the fitness of the homozygous wild-type genotype is 1, the fitness of the heterozygous genotype is 1 + $hs$, and the fitness of the homozygous mutant genotype is 1 + $s$. The first model assumes that $h$ is 0.5 and does not depend on $s$ (additive model). The second model assumes that $h$ is independent of $s$, but differs from 0.5 (constant $h$ model). This model allows for dominant or recessive mutations. The third model assumes a functional relationship between $h$ and $s$ ($h$–$s$ relationship model). We model this relationship with two parameters according to Eq. (1). The first parameter, $\theta_{intercept}$, defines the value of $h$ at $s = 0$. The second parameter, $\theta_{rate}$, defines how quickly $h$ approaches zero with decreasing negative selection coefficient (see Fig. 1d). We assume that $\theta_{rate}$ is positive. Large positive values of $\theta_{rate}$ imply that $f(s)$ quickly approaches $h = 0$, and even slightly deleterious mutations are recessive. Small positive values of $\theta_{rate}$ imply that only strongly deleterious mutations are recessive.

Overall, we assume that the DFE of new mutations (i.e. the distribution of $s$) follows a gamma distribution[35–37]. Thus, the additive model has two DFE parameters (shape and scale of the gamma DFE) and no dominance parameters, since we fix $h$ to be 0.5. The constant $h$ model has one additional parameter, the value of $h$. The $h$–$s$ relationship model has two additional parameters, $\theta_{intercept}$ and $\theta_{rate}$. Note that when $\theta_{rate}$ approaches zero, the $h$–$s$ relationship model of Eq. (1) converges to the constant $h$ model, and when $\theta_{rate}$ approaches zero and $\theta_{intercept}$ approaches 0.5, the model converges to the additive model. Thus, the three models are nested, and we can formulate a likelihood ratio test based on maximum log likelihoods (LL) comparing the three different dominance models. The test statistic $\Lambda$ is defined as $2(LL_{H1}-LL_{H0})$, where H0 is the null hypothesis (either additivity or constant $h$) and H1 is the alternative hypothesis (either constant $h$ or $h$–$s$ relationship). The statistic $\Lambda$ is asymptotically chi-square distributed, with degrees of freedom equal to the difference in the number of parameters between the null and the alternative model. Thus, we formulate three different tests: (1) testing the constant $h$ model (H1) against the additive model (H0). (2) Testing the $h$–$s$ relationship model (H1) against the additive model (H0). (3) Testing the $h$–$s$ relationship model (H1) against the constant $h$ model (H0).

**Inference using a single outcrossing population**. We developed a Poisson Random Field model of polymorphisms[14] for estimating the parameters in the models described above. We assume that nonsynonymous mutations are under the effects of purifying selection, and we assume that synonymous mutations are neutral. We present two approaches to estimate these parameters from the data: (1) estimating dominance using data from a single outcrossing population (e.g. *A. lyrata*), and (2) using data from both an outcrossing (e.g. *A. lyrata*) and a highly inbreeding population (e.g. *A. thaliana*) simultaneously to estimate dominance. We start by presenting the first approach.

To account for the effects of changes in population size on the nonsynonymous SFS that might confound estimates of selection, we first estimate a demographic model using the synonymous SFS[18]. Selection parameters are then estimated conditional on the estimated demographic model. Previous work has shown that this approach leads to unbiased estimates of the selection parameters by controlling for background selection, selective sweeps, and hidden population structure[36,38]. In particular, this controls for the reduction in effective population size due to selfing, and the increased strength of background selection due to the lower effective recombination rate in the selfing species compared to the outcrossing species.

In short, we infer the parameters of a population size change model using the synonymous SFS under the Poisson Random Field framework (see Huber et al.[36] and Kim et al.[38] for details). For both species that we analyzed (*A. lyrata* and *A. thaliana*), a three-epoch model with three discrete size changes fits better to the synonymous SFS than a two-epoch model or a constant population size model (Supplementary Table 1 and Supplementary Fig. 1). Thus, all subsequent inferences use the three-epoch model.

Conditional on the estimated demographic parameters of the three-epoch model, we next use the nonsynonymous SFS to estimate the selection parameters, i.e. the shape and scale parameter of a gamma distributed DFE ($\Theta_{DFE}$), and the rate and intercept parameter of the $h$–$s$ relationship, $\Theta_h = \{\theta_{intercept}, \theta_{rate}\}$. We use the Poisson likelihood to estimate the combined vector of parameters $\{\Theta_{DFE}, \Theta_h\}$. The likelihood was calculated according to Eq. (2). Here, $\Theta_D$ is a vector of demographic parameters, $X_i$ is the count of SNPs with frequency $i$ in the sample (the entries of the SFS), $\theta$ is the population mutation rate, and $n$ is the sample size. We set $\Theta_D$ to the maximum likelihood estimates of the demographic parameters $\widehat{\Theta_D}$, and $\theta$ to the nonsynonymous population scaled mutation rate, $\theta_{NS} = 4N_e\mu L_{NS}$. We estimated $\theta_{NS}$ from $\theta_S$ by accounting for the difference between the synonymous sequence length ($L_S$) and the nonsynonymous sequence length ($L_{NS}$), assuming a multiplier of $L_{NS} = 2.31 \times L_S$[36].

The expected values of $X_i$ refer to the expected entries of the SFS given demography and selection parameters. We used the software $\partial a \partial i$[31] to compute the expected SFS for a two-dimensional grid of 1 million pairs of $N_e s$ and $h$ values on a grid that is exponential in $N_e s$ and linear in $h$ (see 'Cubic spline interpolation to speed up computation of SFS' below). We vary $h$ from zero (completely recessive) to one (completely dominant), and $N_e s$ from $-N_e$ (i.e. lethal) to $-1 \times 10^{-4}$ (effectively neutral). This set of site frequency spectra is then used to calculate the expected SFS for an arbitrary distribution of $N_e s$ and $h$ values. This is done by numerically integrating over the respective spectra weighted by the gamma distribution. Since we assume one $N_e s$ value corresponds to a single $h$ value (Eq. (1)), this is a one-dimensional integration. The numerical integration was done using the 'numpy.trapz' function as implemented in $\partial a \partial i$.

Numerical optimization is used to find the parameters of the DFE and dominance model that maximize the Poisson likelihood (Eq. (2)). For this optimization step, we use the BFGS algorithm as implemented in the 'optimize. fmin_bfgs' function of scipy. To avoid finding local optima, we repeated every estimation approach from 1000 uniformly distributed random starting parameters. Our approach allows us to estimate the parameters of any arbitrary distribution of $N_e s$ values and any arbitrary function that relates $h$ to $s$ (or $N_e s$).

To summarize, our inference of dominance and DFE parameters ($\Theta_h$, $\Theta_{DFE}$) consists of the following steps. (Step 1) Infer the parameters of a demographic model and the effective (ancestral) population size for the outcrossing population. (Step 2) Conditional on the demographic model, compute the expected SFS for a 2D grid of $h$ and $N_e s$ values. (Step 3) Start at a certain vector of dominance and DFE parameters ($\Theta_h$, $\Theta_{DFE}$). Note that the DFE here is defined in units of $s$, not $N_e s$. (Step 4) Compute the DFE in units of $N_e s$ by scaling the DFE from step 3 by the respective ancestral population size. (Step 5) Compute the $h$ value for the grid of $N_e s$ values according to Eq. (1) and the parameters $\Theta_h$. Then use the 2D lookup table generated in step 2 to find the closest SFS for each pair of $h$ and $N_e s$. Integrate those SFS after weighting according to the DFE to find the expected SFS given the DFE and $h$–$s$ relationship. (Step 6) Given the expected and the empirical SFS for the outcrossing population, compute the log likelihood according to Eq. (2). (Step 7) By repeating steps 3–6, the log likelihood can be calculated for an arbitrary set of parameters. Maximum likelihood parameters are computed numerically by maximizing the likelihood using iterative non-linear optimization methods, such as BFGS or Nelder–Mead[39].

The ancestral effective population size in step 4 is calculated from the demographic model. Fitting the demographic model to the synonymous SFS provided an estimate of $\theta_S = 4N_e\mu L_S$ for synonymous sites, where $\mu$ is the neutral per base-pair mutation rate and $L_S$ is the synonymous sequence length. Using this formula, we estimated $N_e$ by setting the neutral mutation rate to $7 \times 10^{-9}$ (ref. [40]). Note that when partitioning our data into different gene categories and estimating the selection parameters for each category separately, we also allow for a different ancestral $N_e$ and demographic estimates in those categories to control for different levels of background selection in different genomic regions[41–44].

Finally, we can compute the likelihood at the maximum likelihood parameter values for the three different dominance models (i.e. additive model, constant $h$ model, and $h$–$s$ relationship model), and compute the likelihood ratio test statistic $\Lambda$, which allows for model comparison.

## Cubic spline interpolation to speed up computation of SFS.

Step 2 in our inference method involves computing a lookup table of one million SFS for a wide range of $1000 \times 1000$ pairs of $N_e s$ and $h$ values. Although each single computation of a SFS is relatively fast, it is computationally expensive to compute the total of one million SFS with $\partial a \partial i$. We sped up this computation by utilizing the fact that the SFS across close $N_e s$ and $h$ values is fairly smooth. Thus, we only compute the expected SFS for a coarse grid of $50 \times 20$ $N_e s$ and $h$ values, and then interpolate the entries of the SFS for a much finer grid of $1000 \times 1000$ $N_e s$ and $h$ values. The interpolation is done using the CubicSpline function of the python package scipy. interpolate. Each frequency of the SFS is interpolated separately in a two-step process: first, each frequency is interpolated for 1000 positions along the $N_e s$ axis, keeping $h$ constant, leading to a grid of $1000 \times 20$ SFS. Then, each frequency is interpolated along the $h$-axis, keeping $N_e s$ constant, leading to the final grid of $1000 \times 1000$ SFS. Examples of the cubic spline interpolation of frequency classes of the SFS along the $N_e s$ and $h$ axes demonstrate that the interpolation works well for a wide range of $h$, $N_e s$, and minor allele frequency (MAF) values (Supplementary Figs. 5 and 6).

## Inference using an outcrossing and a selfing population.

The nonsynonymous SFS for different values of $h$ can be very similar when modifying the selection coefficient accordingly (see Fig. 1a). This suggests that the power for estimating dominance might be small when using only data from a single outcrossing population. This can be seen in Supplementary Fig. 2a, where simulations with $h = 0.5$ (H0) are compared to simulations with a constant $h$ of 0.46 (H1). Such a small difference in $h$ leads to a considerable overlap in the distribution of the likelihood ratio test statistic $\Lambda$ between simulations under H0 and H1, and there is no power to discriminate those two hypotheses.

We propose to increase power for detecting the true dominance model, and improve parameter estimation, by combining data from an outcrossing species with data from a selfing species. To extend our inference to an inbreeding/ outcrossing pair of populations, we need to calculate the likelihood of the parameters given the nonsynonymous SFS of both populations. When the two species are strongly diverged such that they do not share ancestral polymorphisms, the allele frequencies are independent and the likelihood can be computed as the product of the probability of the outcrossing SFS ($SFS_O$) and the probability of the inbreeding SFS ($SFS_I$). The species pair A. thaliana and A. lyrata meets this assumption, since the probability for shared ancestral polymorphisms is negligibly small and allele frequencies are highly uncorrelated[16]. The log-likelihood of the full model can thus be summed according to Eq. (3). The first term of the sum, the log likelihood of the selection parameters ($\Theta_h$ and $\Theta_{DFE}$) given the outcrossing SFS, is computed using the approach developed above for the case of a single outcrossing population. To calculate the log likelihood for the inbreeding SFS (the second term of the right hand side of Eq. (3)), we need to account for the effect of inbreeding on the SFS. For strongly inbred species such as A. thaliana with a selfing rate of at least 97%[45], we assume that the inbreeding coefficient $F$ is effectively 1 (Supplementary Fig. 9). In this case, the diffusion equation model reduces to a scaled additive model. This can be derived from the formulas of the mean and variance of the change in frequency at an allele frequency $p$: $M(p)$ and $V(p)$. In the most general case, with arbitrary inbreeding and dominance, these two quantities are[46]

$$M(p) = sp(1-p)\{(1-F)[h+(1-2h)p] + F\} \tag{8}$$

$$V(p) = p(1-p)(1+F)/(2N) \tag{9}$$

In the case of additive mutations in an outcrossing population ($F = 0$, $h = 0.5$), these quantities become

$$M(p) = sp(1-p)/2 \tag{10}$$

$$V(p) = p(1-p)/(2N) \tag{11}$$

In the case of a highly inbred population with arbitrary dominance ($F = 1$), these quantities become independent of $h$

$$M(p) = sp(1-p) \tag{12}$$

$$V(p) = p(1-p)/(N) \tag{13}$$

The equations for the case of $F = 1$ (Eqs. (12, 13)) is just a scaled version of the equations for additive mutations in an outcrossing population (Eq. (10, 11)), with twice the change in mean allele frequency (Eq. (12)), and twice as much drift (Eq. (13)). This allows us to use the framework of $\partial a \partial i$, developed for outcrossing populations, and apply it to data from highly selfing populations.

We need to take into account the effect of inbreeding on $M(p)$ and $V(p)$ according to Eqs. (12, 13). The effective population size that we estimate with $\partial a \partial i$ based on the synonymous SFS is already taking into account the effect of inbreeding on $V(p)$, since it is the population size that effectively generates the same amount of drift as the standard Wright–Fisher outcrossing model assumed by $\partial a \partial i$ (i.e. Eq. (11)). Next, we multiply $s$ by a factor of 2 to find the effective selection coefficient $s_e$. Finally, we use these effective parameters, $s_e$ and $N_e$, to compute the expected SFS for the highly selfing population using the framework of $\partial a \partial i$.

The full inference of a common set of dominance and DFE parameters ($\Theta_{hs}$, $\Theta_{DFE}$) is similar to the steps outlined above for a single outcrossing population. (Step 1) Infer the parameters of a demographic model and the effective (ancestral) population size for both the inbreeding and the outcrossing populations. This is done independently for the two populations. (Step 2) Conditional on the demographic model of the outcrossing population, compute the expected SFS for a 2D grid of $h$ and $N_e s$ values. For the inbreeding population, compute the expected SFS for a 1D grid of $N_e s$ values, fixing $h$ to 0.5. (Step 3) Start at a certain vector of dominance and DFE parameters ($\Theta_h$, $\Theta_{DFE}$). Note that the DFE here is defined in units of $s$, not $N_e s$. (Step 4) Compute the DFE in units of $N_e s$ by scaling the DFE with the respective population size separately for the inbreeding and the outcrossing population. For a gamma distributed DFE, this amounts in multiplying the scale parameter by $N_e$. (Step 5) For the inbreeding population, additionally scale the DFE from step 3 by a factor of 2 to derive the effective DFE in units of

$N_e s_e$. (Step 6) For the outcrossing population, compute the $h$ value for the grid of $N_e s$ values according to Eq. (1) and the parameters $\Theta_h$. Then use the 2D lookup table generated in step 2 to find the closest SFS for each pair of $h$ and $N_e s$. Integrate those SFS after weighting according to the DFE to find the expected SFS given the DFE and $h$–$s$ relationship. (Step 7) Compute the expected SFS for the inbreeding population by integrating across the 1D lookup table of SFS after weighting each SFS according to the DFE in units of $N_e s_e$. Note that $h$ is fixed to 0.5. (Step 8) Given the expected and the empirical SFS for both the inbreeding and the outcrossing populations, compute the log likelihood according to Eqs. (2) and (3). (Step 9) By repeating steps 4–8, the log likelihood can be calculated for an arbitrary set of parameters. Maximum likelihood parameters are computed numerically by maximizing the likelihood using iterative non-linear optimization methods such as BFGS or Nelder–Mead[39].

**Bootstrapping and testing model parameters**. Our maximum likelihood approach of inferring the DFE and dominance parameters only returns a point estimate, and does not include a measure of the uncertainty of the estimate. Further, since the approach numerically optimizes the likelihood and estimates demographic parameters, numerical errors might lead to a larger uncertainty in parameters than expected based on the shape of the likelihood function. Thus, we follow a non-parametric bootstrapping approach by Poisson resampling both the synonymous and nonsynonymous empirical SFS and re-estimating the demographic and selection parameters for each resampling. From 20 bootstrapped parameters we then compute the standard error and the 95% confidence interval (Fig. 4c). To test for difference in certain parameters between gene categories, we computed a $Z$-score by dividing the difference in the estimate by the estimated standard error of the difference. The $P$-value is then computed based on the standard normal distribution (Fig. 4d).

**Population genetic simulations**. To test our inference procedure, we simulated data using the forward simulation software PReFerSim[47], but changed the source code of the software to allow for an $h$–$s$ relationship according to Eq. (1). We simulate genome-wide data under the three-epoch model, with $\theta_{Synonymous,Inbreeding} = 41{,}800$, $\theta_{Nonsynonymous,Inbreeding} = 96{,}600$, $\theta_{Synonymous,Outcrossing} = 131{,}600$, and $\theta_{Nonsynonymous,Outcrossing} = 304{,}000$. Here, $\theta$ is $4N_e\mu L$, where $L$ is the respective synonymous or nonsynonymous sequence length, $\mu$ is the neutral mutation rate, and $N_e$ is the ancestral population size. Further, we simulated smaller sets of data that reflect the relatively small number of structural genes, with all values of $\theta$ being 10 times smaller. The simulation parameters for the DFE, the demographic model, and the $h$–$s$ relationship are taken from the empirical estimates from the genome-wide data (see Supplementary Tables 1 and 4). However, the simulations are downscaled to a 50-fold smaller population size than estimated to increase the speed of the simulations[36]. After simulating the respective synonymous and nonsynonymous SFS under both inbreeding and outcrossing, we estimate the demographic parameters, the DFE parameters, and the dominance parameters using our method.

We simulated 100 replicates of the following scenarios: First, we simulated under the additive model, assuming the same DFE in both populations. After running the inference, this leads to the null distribution of the test statistic $\Lambda$ (Fig. 2c, d). Second, we simulated under the constant $h$ model. This leads to the distribution of $\Lambda$ under the alternative hypothesis of constant $h$ (Fig. 2c). Finally, we simulated under the $h$–$s$ relationship model. This leads to the distribution of $\Lambda$ under the alternative hypothesis of an $h$–$s$ relationship (Fig. 2d). We find that the null distributions follow closely to the expectations of the asymptotic theory (Supplemental Note 1), and that we can estimate the true parameters of the $h$–$s$ relationship under all simulation scenarios (Supplementary Fig. 10).

**Simulation of theoretical dominance model**. For the simulations in Fig. 5b, c, we simulated 5000 genes with random intercept and scale parameters and computed $h$ and $s$ of potential mutations in each gene. The cost parameter $c$ was fixed to 0.001. The intercept parameter was sampled from a uniform distribution with values ranging from 0.9 to 1, reflecting the fact that most new mutations are effectively neutral[38]. The scale parameter was sampled from the absolute values of a normal distribution with mean and standard deviation of 0.1, leading to variation in the levels of optimal gene expression that is slightly skewed to lower values (i.e. assuming more genes with small optimal gene expression than with large optimal gene expression).

**Code availability**. The code used for analysis is available at www.github.com/LohmuellerLab/dominance.

**Data availability**. The datasets analyzed during the current study are available in the European Nucleotide Archive (https://www.ebi.ac.uk/ena) using accession numbers PRJEB19780 and PRJNA284572.

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

## Acknowledgements

We thank Dan Balick and Joachim Hermisson for detailed comments on the manuscript. We thank Vincent Castric, Jeff Ross-Ibarra, Stephen Wright, and their respective labs for sharing their discussions on a preprint of this work. This work was supported by a Searle Scholars Fellowship and NIH Grant R35GM119856 (to K.E.L.) and NSF Graduate Research Fellowship DGE-1650604 (to A.D.).

## Author contributions

K.E.L. conceived of and supervised the study. C.D.H. developed the inference method, analyzed the data, and developed the theoretical dominance model. A.D. performed bioinformatics analysis. A.M.H. contributed sequence data. C.D.H., K.E.L, A.D., and A.M.H. wrote the manuscript.

## Additional information

**Competing interests:** The authors declare no competing interests.

