## [Peer Review File · Nature Communications]

Reviewers' comments:

Reviewer #1 (Remarks to the Author):

Overall, this is an impressive study. The fact that h declines when s increases is well-known; however, the authors present a new, sound, and powerful way of estimating their joint distribution which uses data on a pair of related populations - a selfer and an outcrosser. This part of the paper is independent of any evolution of dominance consideration - but the refined Haldane's model developed later in the manuscript is also interesting and makes sense. I recommend publication.

My key point is concerned with what happens to h when $s > 0$ (theta-intercept). It seems that the authors assume that it is equal to 1 - so that almost-neutral deleterious alleles are completely dominant, with homozygotes not being inferior to heterozygotes. I find this situation very unlikely and wonder why did not the authors assume that theta-intercept = 0.5.

Indeed, there may be two reasons for a mutation to possess a small s . First, it may affect the gene severely, but the function of this gene is not very important. In this case $h < 0.5$ (partial recessivity) is likely - due to Fisher, or Wright, or Haldane - they all predict the same thing here - homozygous loss-of-function allele has more than twice the impact of heterozygous LoF allele.

Second, a mutation may affect the function of the gene (crucial or only marginally important) slightly - and in this case a small s must be accompanied by $h = 0.5$ simply because small effects are always additive - whatever equations describe the corresponding dynamics, they can be expanded into Taylor series.

Thus, I cannot see any mechanism for $h > 0.5$ being common. What would happen in $h > 0.5$ when $s > 0$ in Fig. 2B. I could not believe that a fit would become worse! And if it does, it would be important to understand why. A priori theta-intercept = 0.5 (or less) looks much more likely.

Small points:

1. Strictly speaking, only Fisher's model assumes evolution of dominance - in other cases, something else is evolving (by selection), and dominance changes only as a by-product.
2. Are you sure that Fisher's model does not predict the same negative relationship between s and h ? Alleged selection for dominance modifiers should be stronger when the gene is more important (not that I believe in such selection).
3. Define SFS when it is used for the 1st time.

Alex Kondrashov (I sign all my reviews)

Reviewer #2 (Remarks to the Author):

Huber et al. write a thought provoking piece in which they seek to deconvolute the relationship between dominance and selection, offering a new dynamic model for the relationship between selection and dominance to predict expression levels. Using data from one each of a selfing and outcrossing Arabidopsis species enables the authors to co-estimate the distribution of fitness effects and of dominance for nonsynonymous mutations. The authors have corroborated their findings and model predictions with examples from other organisms (i.e., some support from yeast and fly experiments for their observations).

In general, I think the manuscript offers a new look at an old debate in the field and is likely to be of very broad interest. There are some assumptions the authors make that I would like them to reexamine, although perhaps doing so is well beyond the scope of this manuscript. For example, why assume that the cost per unit gene expression is the same for each gene? And why assume that genes are always optimally expressed? I think readers need more context to understand why they should accept the latter as true, at the very least.

I think the model testing frameworks developed are quite nice and are rigorously applied. My other comments are largely minor, except to say the manuscript is very very dense, due to its short length. In particular, some of the quantitative methods used to infer dominance should be moved under results. These approaches are surely part of the contribution this manuscript is making to the field.

Typos/minor comments:

line 63: change "this relationship" to "this negative relationship"

line 76: estimate is misspelled

line 82: change "our empirical patterns" to "the empirical patterns we observe"

line 132: change "that of the null data" to "under the constant $h=0.5$ model"

line 202: putting "cost" in the equation looks at first glance like some cosine variant...typesetting might fix this but I would suggest parametrizing cost as c .

line 225: why is positive in parens?

Reviewer #3 (Remarks to the Author):

This paper takes a clever approach to a very challenging problem: the joint distribution of h and s in natural populations. The authors then articulate a model for the genetics of dominance and show that their data fit it well.

The inference method here compares the site frequency spectra between outcrossing and predominantly selfing species of Arabidopsis. There are an enormous number of assumptions here, and the authors do an impressive job of anticipating and addressing concerns about these assumptions (e.g., use of PRF in the face of population structure and selection at linked sites). Nevertheless, I have some residual anxieties.

Initially the authors test three models: additivity, constant $h \neq 0.5$, and $h = f(s)$. The first two of these are really strawmen, so the main interest is the exact shape of the relationship between h and s . Here, I find the results a bit surprising— mutations with s less than -0.001 are extremely recessive. My intuition (based on nothing, to be sure) is that mutations of that order should be weakly recessive, while $s > 0.1$ or so should be strongly recessive. I think human GWAS results say this too: for example, height's under selection in many populations, and there are 100s of known additive height variants. So what are the key assumptions that lead the authors to this result? I think it's mostly the functional form of $h = f(s)$, methods eq. 1 at line 339. The authors do not devote any space to justifying this function over alternatives. A particularly puzzling thing to me is that they consistently find the intercept to be 1 rather than 0.5. I can't think of a good reason that the smallest-effect class should be dominant, nor is that a prediction of e.g., Kascser & Burns or other model of dominance that I know of, nor of the authors' own model (e.g., Fig 4 B and C). Why should there be an effect size across which the average mutation flips from being recessive to being dominant? The fact that the model consistently estimates theta-intercept = close to 1 suggests to me that the function is a bad model of the biology, and I'd like the authors to do more to convince me that it's right and that I should trust the parameters they estimate. If I don't trust theta-intercept, I can't trust theta-rate.

(One reason for the authors' choice may be that it permits use of likelihood ratio tests, as additivity and constant h are nested. But I don't think this is a great reason, particularly because really the nested models just constrain some parameters at their boundaries, in which case the test statistic isn't really chi-squared on the difference in parameters; that's why the authors have to use simulations to get p-values anyway.)

Notwithstanding my concern about eq. 1, the rest of the data analysis is exceptionally clever and the authors provide a clear and convincing exposition of it. My only other substantive questions about it are:

- 1) How are the inferences affected by variance in the h,s relationship? That is, mutations with a given s certainly draw from a distribution of h . Does that matter? Will the high- h tail of mutations for a given s dominate the population genetic outcomes?
- 2) How reasonable is the assumption that the two species share a DFE? The transition to selfing should cause relaxed selection on diverse aspects of outcrossing function, so that lots of mutations that are deleterious in *lyrata* should be neutral in *thaliana*, independent of population size.

Note that I don't doubt at all that the data are robust to these questions with regard to the choice of $h=f(s)$ over any constant h model; the issue is the robustness of the specific details of $h=f(s)$.

One of the motivations for the research is to test existing evolutionary models of dominance. I'd say that Fisher's model has been dead for about 50 years, and certainly since Orr 1991, so there's no need to spend a lot of time on it. The authors argue that the presence of a relationship between h and s across gene categories implies that the metabolic theory can't be the whole story. They extend an existing model to incorporate variation among genes in optimal expression levels, showing that it predicts the relationships they observe. This is an interesting and valuable addition to the dominance theory literature.

Minor things:

I found the reference to the omnigenic model at line 287 to be mysterious; I don't see how the authors's results relate to that model.

Line 475: when two species do not share ancestral polymorphism.... Add here that *A. lyrata* and *A. thaliana* fit this description. Most readers will not already know this.

Fig 1 A and B: I spent quite a while confused by these figures until I realized that each set of bars represents a totally different unspecified DFE. Maybe there's a way to say that in the in-figure key.

Response to Reviewers:

Reviewer #1 (Remarks to the Author):

Overall, this is an impressive study. The fact that h declines when s increases is well-known; however, the authors present a new, sound, and powerful way of estimating their joint distribution which uses data on a pair of related populations - a selfer and an outcrosser. This part of the paper is independent of any evolution of dominance consideration - but the refined Haldane's model developed later in the manuscript is also interesting and makes sense. I recommend publication.

We thank Reviewer 1 for the positive comments and for acknowledging the novelty and strength of our approach.

My key point is concerned with what happens to h when $s > 0$ (theta-intercept). It seems that the authors assume that it is equal to 1 - so that almost-neutral deleterious alleles are completely dominant, with homozygotes not being inferior to heterozygotes. I find this situation very unlikely and wonder why did not the authors assume that theta-intercept = 0.5.

Indeed, there may be two reasons for a mutation to possess a small s . First, it may affect the gene severely, but the function of this gene is not very important. In this case $h < 0.5$ (partial recessivity) is likely - due to Fisher, or Wright, or Haldane - they all predict the same thing here - homozygous loss-of-function allele has more than twice the impact of heterozygous LoF allele.

Second, a mutation may affect the function of the gene (crucial or only marginally important) slightly - and in this case a small s must be accompanied by $h = 0.5$ simply because small effects are always additive - whatever equations describe the corresponding dynamics, they can be expanded into Taylor series.

Thus, I cannot see any mechanism for $h > 0.5$ being common. What would happen in $h > 0.5$ when $s > 0$ in Fig. 2B. I could not believe that a fit would become worse! And if it does, it would be important to understand why. A priori theta-intercept = 0.5 (or less) looks much more likely.

We thank Reviewer 1 for suggesting the need to clarify and further investigate the dominance coefficient of almost-neutral deleterious mutations. When we inferred the relationship between h - s , we did not constrain our inference of the dominance coefficient of almost-neutral mutations to be close to 0.5, but rather allowed the intercept of our h - s relationship function (theta_intercept) to vary freely between 0 and 1. We consistently estimated theta_intercept to be close to one. Interestingly, while it may seem biologically unlikely for almost neutral deleterious mutations to have $h > 0.5$, we note that experimental results in yeast, using entirely different approaches and patterns in the data, also estimated $h > 0.5$ for nearly neutral mutations (Phadnis and Fry 2005; Agrawal and Whitlock 2011). Thus, there is some precedent for very nearly neutral mutations to be dominant.

However, in the revised manuscript we now investigate a number of other factors that might lead to inference of $h > 0.5$. First, we tested the effect of misspecification of the distribution of fitness effects (DFE). Recent work in humans has suggested that the true DFE might contain more effectively neutral mutations than can be accounted for by a gamma distribution (Kim et al.

2017). To do this, we added a neutral point mass to the gamma distributed DFE and re-inferred the DFE and h - s relationship from the *Arabidopsis* data (Supplementary Table 3). However, allowing for this more complex DFE does not push the estimate of $\theta_{\text{intercept}}$ to lower values.

Second, we now test a more flexible shape of the h - s relationship function than the inverse relationship function ($h = f[s] = 1/(1/\theta_{\text{intercept}}) - \theta_{\text{rate}}*s$) that we used before. We now also test a logistic function that has one additional parameter (θ_{offset}) to allow for a flattening of the h - s relationship function for values of s close to zero ($h = f[s] = \theta_{\text{intercept}}*(1+\exp(-\theta_{\text{offset}}))/(1+\exp(\theta_{\text{rate}}*\text{abs}(s)-\theta_{\text{offset}}))$). However, when we fit this more flexible h - s relationship to the data, the dominance coefficients of almost-neutral mutations are still inferred to be close to 1 (see our new Fig. 3a and Supplementary Table 3).

Third, our inference model assumes that there is no variation in the dominance coefficient h conditional on a given selection coefficient s , i.e. the dominance coefficient h follows the h - s relationship function and there is no additional (unexplained) variation in h . In reality, this assumption might be violated and there might be a distribution of h values instead of just a single value of h for a given s . To test the robustness of our inference approach to a distribution of h , we simulated data under a model that assumes additional variance in h values sampled from a beta distribution with a fixed standard deviation of 0.1. This model generates considerable additional variation in h (see Fig. 3b). We further assume that the true $\theta_{\text{intercept}}$ is 0.5. In other words, on average almost-neutral mutations are truly additive. When simulating data under this model and estimating the h - s relationship parameters from the simulated data using our composite likelihood approach, we find that the estimated curves fall well within the point cloud of h and s values and fairly well reflect the increase in recessivity with increasing deleteriousness of mutations. However, we do find that our estimates of the $\theta_{\text{intercept}}$ parameter are biased to larger values than the true expected value of 0.5. Thus, the fact that we estimate a $\theta_{\text{intercept}}$ parameter of close to one in our *Arabidopsis* data might reflect a large variation in the dominance coefficient for almost-neutral mutations, even though on average those mutations might actually be additive. Thus, we cannot conclude from our results that almost neutral mutations are dominant.

To account for the fact that variation in h might bias $\theta_{\text{intercept}}$ to larger values, we constrained both inverse and logistic h - s relationship functions to a $\theta_{\text{intercept}}$ of 0.5 such that almost-neutral mutations are additive. We found that this constrained functional relationship still fits significantly better to the data than the constant h model, but not better than the unconstrained model (Supplementary Table 3). Thus, even when we constrain $\theta_{\text{intercept}}$ to the biologically more reasonable value of 0.5, we still observe strong support for a negative relationship between h and s , consistent with our original conclusions. Additionally, for deleterious mutations with $s < -0.0005$ the inference of the mean dominance coefficient is fairly robust to different assumptions about almost-neutral mutations (Fig. 3a).

In sum, misspecification of the exact shape of the h - s relationship leads to some uncertainty in the exact mean value of the dominance coefficient of almost-neutral mutations (see new Fig. 3a). However, for deleterious mutations with $s < -0.0005$ the inference of the mean dominance coefficient is accurate, and is not affected by the bias in the $\theta_{\text{intercept}}$.

We changed our manuscript to reflect this refined interpretation of our inference on lines 145-154 where we now write:

Lastly, we tested the effect of assuming alternative functions for the h - s relationship. In particular, we tested 1) a logistic function, and 2) the effect of constraining $\theta_{\text{intercept}}$ to a value of 0.5, i.e. assuming that almost-neutral mutations are additive. Although assuming different functional relationships between h and s introduces some uncertainty in the inferred dominance coefficient for mutations with a selection coefficients between -5×10^{-4} and 0, the inference of the dominance coefficient for deleterious mutations with $s < -5 \times 10^{-4}$ is robust to the assumed functional form (Fig. 3a). Notably, irrespective of the functional form, we observe strong statistical support for a negative relationship between h and s (Supplementary Table 3). Further, this result is robust to assuming a DFE that allows for a proportion of neutral mutations (DFE & neutral in Fig. 3a).

And on lines 174-187:

Our inference model assumes no variation in the dominance coefficient h conditional on a given selection coefficient s . In reality, for any given s , there might be a distribution of h values instead of a single value of s . To test how this additional variation in h might affect our inference, we simulated data under a model that assumes additional variance in h by sampling h from a beta distribution with a fixed standard deviation of 0.1 (see Fig. 3b). We further assume that the true $\theta_{\text{intercept}}$ is 0.5. When estimating the h - s relationship parameters from the simulated data, we find that the estimated curves fall well within the point cloud of h and s values and reflect the increase in recessivity with increasing deleteriousness of mutations fairly well. This result suggests that our estimates of the mean h - s relationship for moderately deleterious mutations are robust to variation in h . However, the estimated $\theta_{\text{intercept}}$ parameter is biased to larger values than the true expected value of 0.5. Thus, our estimates of $\theta_{\text{intercept}}$ close to one in the *Arabidopsis* data might reflect a large variance in the dominance coefficient for almost-neutral mutations, even though on average those mutations might be additive. Therefore, we cannot conclude from our results that almost neutral mutations are mostly dominant.

Small points:

1. Strictly speaking, only Fisher's model assumes evolution of dominance - in other cases, something else is evolving (by selection), and dominance changes only as a by-product.

We thank reviewer 1 for making us aware of this issue. We agree that in general, 'evolution of dominance' should be reserved for models where dominance relationships have been molded by natural selection to some extent and thus should not be used as a general term for all dominance models. We changed the wording at several positions in the manuscript to reflect this (see below).

However, we believe it is accurate to use 'evolution of dominance' when referring to our model of dominance. Our model assumes that evolution acts on a 'modifier' locus (the regulatory sequence) to optimize gene expression, which leads to a change in the dominance of mutations at a second locus (the gene). It thus appears to us that it is, at least conceptually, not drastically different from Fisher's two-locus evolutionary model of dominance. However, we removed the term 'evolution of dominance' from the following lines of the revised manuscript:

Line 67-70: While population genetic approaches to estimate the degree of dominance from segregating genetic variation exist^{14,15}, they have not been widely applied to empirical data nor have they been used to test models regarding the mechanistic basis of dominance.

Line 80-81: We then explore which mechanistic models of dominance can explain key biological properties in our data.

Line 197-198: We next sought to test which theoretical model of the basis of dominance can explain our data.

Line 326-329: Further, we find that existing models of the mechanistic basis of dominance, such as Fisher's modifier model⁵, the metabolic theory model^{6,7}, or newer models based on fitness landscapes²¹ do not explain patterns of dominance that we see across all types of genes.

2. Are you sure that Fisher's model does not predict the same negative relationship between s and h ? Alleged selection for dominance modifiers should be stronger when the gene is more important (not that I believe in such selection).

According to previous literature, Fisher's model does not predict a negative relationship between s and h (Charlesworth 1979). Charlesworth shows that the intensity of selection on a dominance modifier depends only on the mutation rate and the extent to which the level of dominance is affected by the modifier, but is independent of the homozygous effect of the deleterious mutation (i.e. the selection coefficient s). The reason is that the frequency of heterozygotes in more important genes (more deleterious s) is lower than in less important genes (less deleterious s). The lower frequency of heterozygotes in more important genes cancels the larger effect of the modifier mutation on fitness. Thus, under Fisher's model, dominance should be reduced at the same rate for genes with drastic effects on fitness as for genes with minor effects on fitness, so that no relationship between s and h should exist (Charlesworth 1979). We now cite this paper at the relevant positions in our manuscript (line 62 and 199). We are not aware of any formulation of Fisher's model that would predict that a relationship between h and s should evolve.

3. Define SFS when it is used for the 1st time.

We added an explanatory sentence when we first use SFS in the main text (line 91-92):

We use the distribution of allele frequencies in a sample, or site frequency spectrum (SFS), as summary of genetic variation in a population.

Reviewer #2 (Remarks to the Author):

Huber et al. write a thought provoking piece in which they seek to deconvolute the relationship between dominance and selection, offering a new dynamic model for the relationship between selection and dominance to predict expression levels. Using data from one each of a selfing and outcrossing Arabidopsis species enables the authors to co-estimate the distribution of fitness effects and of dominance for nonsynonymous mutations. The authors have corroborated their findings and model predictions with examples from other organisms (i.e., some support from yeast and fly experiments for their observations).

In general, I think the manuscript offers a new look at an old debate in the field and is likely to be of very broad interest. There are some assumptions the authors make that I would like them to reexamine, although perhaps doing so is well beyond the scope of this manuscript. For example, why assume that the cost per unit gene expression is the same for each gene?

And why assume that genes are always optimally expressed? I think readers need more context to understand why they should accept the latter as true, at the very least.

We thank Reviewer 2 for his/her positive comments on our manuscript. We also appreciate the request to more carefully examine the sensitivity of our model of the evolution of dominance to certain assumptions. Generally, our theoretical model of dominance recapitulates two key features seen in the empirical data analysis: First, more strongly deleterious mutations are more recessive than less deleterious mutations, and second, mutations become slightly less recessive in genes with high expression level. We found that these two predictions are highly robust to an order of magnitude change in the cost of gene expression (see Supplementary Fig. 12). We have added a figure to the revised manuscript (Supplementary Fig. 13) that shows that the same two predictions are expected when gene expression of the wild-type genotype is not optimal but at 80% of the optimal expression level (i.e. sub-optimal expression level). We added comments on the robustness of our theoretical model in lines 255-256 of the revised manuscript:

However, key features of the model are highly robust to different values of c such that variation in c between genes should not affect our conclusions (Supplementary Fig. 12 and 13).

And on line 300-302:

Suboptimal gene expression, where the expression level is at 80% of its optimal value, leads to qualitatively similar behaviour as when assuming optimal gene expression (Supplementary Fig. 13).

Further, we added a citation on line 284-286 to a comparative study in *Drosophila* that suggests that selection is highly effective in keeping gene expression close to its optimal value (Bedford and Hartl 2009), suggesting that gene expression is likely close to the optimal expression level:

Theoretical and empirical arguments suggest that selection should be highly effective in keeping gene expression close to its optimal value²⁴, although we also study the model under suboptimal expression (see below).

I think the model testing frameworks developed are quite nice and are rigorously applied. My other comments are largely minor, except to say the manuscript is very very dense, due to its short length. In particular, some of the quantitative methods used to infer dominance should be moved under results. These approaches are surely part of the contribution this manuscript is making to the field.

We now moved some of the details of our dominance inference from methods into results and hope that this makes the results more readable. In particular, we added the lines 108-136:

We model the relationship between s and h according to equation (1):

$$h = f(s) = \frac{1}{\frac{1}{\theta_{\text{intercept}}} - \theta_{\text{rate}}s}, \quad (1)$$

where $\theta_{\text{intercept}}$ defines the value of h at $s = 0$ and θ_{rate} determines how quickly h approaches zero with decreasing negative selection coefficient (see Fig. 1d). We then extended the Poisson random-field model of polymorphisms¹⁴ for estimating the two parameters of this relationship, $\theta_{\text{intercept}}$ and θ_{rate} (see Methods for further details). To account for the effects of changes in

population size on the nonsynonymous SFS that might confound estimates of selection, we first estimate a three-epoch demographic model using the synonymous SFS¹⁸. The shape and scale parameter of a gamma distributed DFE (Θ_{DFE}) and the rate and intercept parameter of the h - s relationship, $\Theta_h = \{\theta_{intercept}, \theta_{rate}\}$, are then estimated conditional on the estimated demographic model. We then use the Poisson likelihood to estimate the combined vector of parameters $\{\Theta_{DFE}, \Theta_h\}$ according to equation (2):

$$L(\Theta_{DFE}, \Theta_h | \Theta_D, \theta, X_i) = \prod_{i=1}^{n-1} \frac{E[X_i | \Theta_D, \Theta_{DFE}, \Theta_h, \theta]^{X_i}}{X_i!} e^{-E[X_i | \Theta_D, \Theta_{DFE}, \Theta_h, \theta]} \quad (2)$$

Here, Θ_D is a vector of demographic parameters, X_i is the count of SNPs with frequency i in the sample (the entries of the SFS), θ is the population mutation rate, and n is the sample size. To combine data from the outcrossing species *A. lyrata* with data from the selfing species *A. thaliana*, we compute the combined log-likelihoods (LL) over both datasets by summing over the individual log-likelihoods:

$$LL(\Theta_h, \Theta_{DFE} | SFS_O, SFS_I, \Theta_{D,I}, \theta_I, \Theta_{D,O}, \theta_O) = LL_O(\Theta_h, \Theta_{DFE} | SFS_O, \Theta_{D,O}, \theta_O) + LL_I(\Theta_h, \Theta_{DFE} | SFS_I, \Theta_{D,I}, \theta_I) \quad (3)$$

Finally, we infer the maximum likelihood parameter values for three different dominance models (i.e. additive model, constant h model, and h - s relationship model; see Fig. 1d), and compute the likelihood ratio test statistic Λ to compare between the models (see Methods for further details).

Typos/minor comments:

line 63: change "this relationship" to "this negative relationship"

We changed this line accordingly.

line 76: estimate is misspelled

We fixed this typo.

line 82: change "our empirical patterns" to "the empirical patterns we observe"

We changed this line accordingly.

line 132: change "that of the null data" to "under the constant $h=0.5$ model"

We changed this line accordingly.

line 202: putting "cost" in the equation looks at first glance like some cosine variant....typesetting might fix this but I would suggest parametrizing cost as c .

We changed the parameterization of the fitness cost of gene expression to c .

line 225: why is positive in parens?

We removed the parentheses.

Reviewer #3 (Remarks to the Author):

This paper takes a clever approach to a very challenging problem: the joint distribution of h and s in natural populations. The authors then articulate a model for the genetics of dominance and show that their data fit it well.

We thank the reviewer for the positive comments on our overall approach and analysis.

The inference method here compares the site frequency spectra between outcrossing and predominantly selfing species of *Arabidopsis*. There are an enormous number of assumptions here, and the authors do an impressive job of anticipating and addressing concerns about these assumptions (e.g., use of PRF in the face of population structure and selection at linked sites). Nevertheless, I have some residual anxieties.

Initially the authors test three models: additivity, constant $h \neq 0.5$, and $h = f(s)$. The first two of these are really strawmen, so the main interest is the exact shape of the relationship between h and s . Here, I find the results a bit surprising— mutations with s less than -0.001 are extremely recessive. My intuition (based on nothing, to be sure) is that mutations of that order should be weakly recessive, while $s > 0.1$ or so should be strongly recessive. I think human GWAS results say this too: for example, height's under selection in many populations, and there are 100s of known additive height variants. So what are the key assumptions that lead the authors to this result? I think it's mostly the functional form of $h = f(s)$, methods eq. 1 at line 339. The authors do not devote any space to justifying this function over alternatives.

We thank Reviewer 3 for appreciating our careful investigation of the robustness of our inference approach. In the revised manuscript, we now also test a range of possible functional forms of $h = f(s)$ to investigate how our inference of the h - s relationship is affected by the choice of this function (see the new Fig. 3a). In particular, we tested a logistic function that has one additional parameter (h_offset) to allow for a flattening of the h - s relationship function for values of s close to zero ($h = f[s] = \text{theta_intercept} * (1 + \exp(-\text{theta_offset})) / (1 + \exp(\text{theta_rate} * \text{abs}(s) - \text{theta_offset}))$). We also test how the estimated relationship changes when theta_intercept is fixed to a value of 0.5, i.e. when almost-neutral mutations are assumed to be additive. Although there is some uncertainty of the dominance coefficient of mutations with selection coefficient between $-5e-4$ and 0, regardless of which functional form is fit to the data, mutations with s more deleterious than -0.001 are fairly recessive.

We do not believe that our present findings conflict with GWAS results for height. First, since height is a highly complex trait, we do not expect selection on each individual causal allele to be particularly strong, and the majority of detected variants should have a selection coefficient less than -0.001 . See for example the landmark study of Turchin et al 2012, where they estimate widespread weak selection on height-associated SNPs, with s between only $-1e-5$ and $-1e-3$. Thus, their results are still compatible with more strongly deleterious mutations being recessive. Second, even though height-associated variants might have an additive effect on height, their effect on fitness can be non-additive because of the curvature of the fitness function for traits under stabilizing selection. Thus, the fact that variants affecting height might be predominantly additive in their effect on height does not directly allow us to infer that they should also have an additive effect on fitness. A QTL mapping study on the contribution of alleles to variation in traits that are more directly related to fitness, in inbred lines of maize, suggests that indeed more damaging variants are usually more recessive (Yang et al. 2017).

Lastly, while we agree the particular form of the h - s relationship is of paramount importance, we believe there is value to rejecting the additive and fully recessive models. Previous molecular population genetic studies of amino acid changing mutations have always assumed that $h=0.5$ (i.e. additivity). While this has typically been done for mathematical convenience, it has remained an open question whether slightly deleterious mutations were additive or not. Experimental studies have focused on strongly deleterious mutations and have little power and may suffer from biases for slightly deleterious mutations. Thus, our study is the first using genetic variation data from natural populations to show that the h - s relationship model better fits the data than the additive or fully recessive models.

A particularly puzzling thing to me is that they consistently find the intercept to be 1 rather than 0.5. I can't think of a good reason that the smallest-effect class should be dominant, nor is that a prediction of e.g., Kascser & Burns or other model of dominance that I know of, nor of the authors' own model (e.g., Fig 4 B and C). Why should there be an effect size across which the average mutation flips from being recessive to being dominant? The fact that the model consistently estimates theta-intercept = close to 1 suggests to me that the function is a bad model of the biology, and I'd like the authors to do more to convince me that it's right and that I should trust the parameters they estimate. If I don't trust theta-intercept, I can't trust theta-rate.

We thank Reviewer 3 for pushing us to further investigate the intercept of our h - s relationship. Indeed, we consistently estimated the intercept to be close to one. We note that experimental results in yeast, using entirely different approaches and patterns in the data, also estimated $h>0.5$ for nearly neutral mutations (Phadnis and Fry 2005; Agrawal and Whitlock 2011). While there is some precedent for very nearly neutral mutations to be dominant, we agree that it is biologically more reasonable to assume that almost-neutral mutations are additive.

As described above in response to Reviewer 1, in the revised manuscript we now also test a logistic function that has one additional parameter (θ_{offset}) to allow for a flattening of the h - s relationship function for values of s close to zero. However, when we fit this more flexible h - s relationship to the data, our estimates of the dominance coefficients of almost-neutral mutations are close to 1 (see our new Fig. 3a). These results are also presented in lines 145-154 of the revised manuscript.

Thus, we tested another assumption that we make in our inference: that there is no variation in the dominance coefficient h conditional on a given selection coefficient s , i.e. the dominance coefficients h follows the h - s relationship function and there is no additional (unexplained) variation in h . In reality, this assumption might be violated and there might be a distribution of h values instead of just a single value of h for a given s . To test the robustness of our inference approach, we simulated data under a model that assumes additional variance in h values sampled from a beta distribution with a fixed standard deviation of 0.1. This model generates considerable additional variation in h (see Fig. 3b). We further assume in the simulations that $\theta_{\text{intercept}}$ is 0.5. In other words, on average almost-neutral mutations are truly additive. When simulating data under this model and estimating the h - s relationship parameters from the simulated data using our composite likelihood approach, we find that the estimated curves fall well within the point cloud of h and s values and fairly well reflect the increase in recessivity with increasing deleteriousness of mutations. However, we do find that the estimates of the $\theta_{\text{intercept}}$ parameter are biased to larger values than the true expected value of 0.5. Thus, these results suggest that one should not literally interpret the $\theta_{\text{intercept}}$ parameter. Our results are also consistent with most almost neutral mutations being additive on average, but with considerable variation in the dominance coefficient.

Importantly, despite the fact that there is some bias in the estimates of the $\theta_{\text{intercept}}$ parameter, for deleterious mutations with $s < -0.0005$, our estimates of the mean dominance coefficient are accurate, and are not affected by the bias in the $h_{\text{intercept}}$. Further, even when we constrain $\theta_{\text{intercept}}$ to the biologically more reasonable value of 0.5 we still observe strong support for a negative relationship between h and s . Moreover, for deleterious mutations with $s < -0.0005$ the inference of the mean dominance coefficient is fairly robust to different assumptions about almost-neutral mutations (Fig. 3a). We updated our manuscript to reflect this more refined interpretation of our inference on lines 145-154:

Lastly, we tested the effect of assuming alternative functions for the h - s relationship. In particular, we tested 1) a logistic function, and 2) the effect of constraining $\theta_{\text{intercept}}$ to a value of 0.5, i.e. assuming that almost-neutral mutations are additive. Although assuming different functional relationships between h and s introduces some uncertainty in the inferred dominance coefficient for mutations with a selection coefficients between -5×10^{-4} and 0, the inference of the dominance coefficient for deleterious mutations with $s < -5 \times 10^{-4}$ is robust to the assumed functional form (Fig. 3a). Notably, irrespective of the functional form, we observe strong statistical support for a negative relationship between h and s (Supplementary Table 3). Further, this result is robust to assuming a DFE that allows for a proportion of neutral mutations (DFE & neutral in Fig. 3a).

(One reason for the authors' choice may be that it permits use of likelihood ratio tests, as additivity and constant h are nested. But I don't think this is a great reason, particularly because really the nested models just constrain some parameters at their boundaries, in which case the test statistic isn't really chi-squared on the difference in parameters; that's why the authors have to use simulations to get p-values anyway.)

We agree, since we use simulations to get p-values there is no requirement for hypotheses to be nested.

Notwithstanding my concern about eq. 1, the rest of the data analysis is exceptionally clever and the authors provide a clear and convincing exposition of it. My only other substantive questions about it are:

1) How are the inferences affected by variance in the h , s relationship? That is, mutations with a given s certainly draw from a distribution of h . Does that matter? Will the high- h tail of mutations for a given s dominate the population genetic outcomes?

We thank Reviewer 3 for raising this important issue. As outlined above as well as in our response to Reviewer 1, we now in the revised manuscript explicitly test the performance of our method under these conditions. Specifically, we added simulations from a model where mutations with a given s are drawn from a distribution of h values. Even though we simulate considerable variation in h , assuming a beta-distributed deviation from the mean h - s relationship ($sd = 0.1$, see the new Fig. 3b), we found that our composite likelihood approach assuming a single value of h for each s value still captured the mean relationship between h and s fairly well. However, the results suggest that a high- h tail of almost-neutral mutations might be responsible for a bias of the intercept parameter to larger values than the simulated mean h of 0.5 for those mutations. Thus, our approach is robust to unmodeled variance in the distribution of h . We now present these results in the revised manuscript in lines 174-187:

Our inference model assumes no variation in the dominance coefficient h conditional on a given selection coefficient s . In reality, for any given s , there might be a distribution of h values instead

of a single value of s . To test how this additional variation in h might affect our inference, we simulated data under a model that assumes additional variance in h by sampling h from a beta distribution with a fixed standard deviation of 0.1 (see Fig. 3b). We further assume that the true $\theta_{\text{intercept}}$ is 0.5. When estimating the h - s relationship parameters from the simulated data, we find that the estimated curves fall well within the point cloud of h and s values and reflect the increase in recessivity with increasing deleteriousness of mutations fairly well. This result suggests that our estimates of the mean h - s relationship for moderately deleterious mutations are robust to variation in h . However, the estimated $\theta_{\text{intercept}}$ parameter is biased to larger values than the true expected value of 0.5. Thus, our estimates of $\theta_{\text{intercept}}$ close to one in the *Arabidopsis* data might reflect a large variance in the dominance coefficient for almost-neutral mutations, even though on average those mutations might be additive. Therefore, we cannot conclude from our results that almost neutral mutations are mostly dominant.

2) How reasonable is the assumption that the two species share a DFE? The transition to selfing should cause relaxed selection on diverse aspects of outcrossing function, so that lots of mutations that are deleterious in *lyrata* should be neutral in *thaliana*, independent of population size.

We thank reviewer 3 for suggesting the need to clarify the effect of selfing on the DFE. We want to point out that we only assume that the distribution of selection coefficients (the DFE on the scale of s) is the same between *A. lyrata* and *A. thaliana*, i.e. there is no systematic change in the fitness effect of mutations on a genome-wide level. However, we test how much our inference might be influenced by such a genome-wide change in the DFE between *A. thaliana* and *A. lyrata*. When we base our inference solely on the outcrossing *A. lyrata* data, not making any assumption about the DFE in *thaliana*, we still see significant support for the h - s relationship and similar parameter estimates compared to using data from both species simultaneously (see Supplementary Table 2). This result suggests that our inferences are not biased by differing DFEs on the scale of s between species.

Further, our approach does control for the increase in drift that is caused by the transition to selfing by estimating the demographic model from the synonymous SFS. Our approach controls for the reduction in effective population size by a factor of 2 due to selfing. It also controls for a further reduction in effective population size caused by background selection, as high rates of selfing strongly reduce the effective recombination rate and thus increase the effect of background selection. Previous work has shown that our approach effectively controls for the confounding effect of drift and background selection (Huber et al. 2017). We added more clarification of this on line 425-429:

Previous work has shown that this approach leads to unbiased estimates of the selection parameters by controlling for background selection, selective sweeps, and hidden population structure^{36,38}. In particular, this controls for the reduction in effective population size due to selfing, and the increased strength of background selection due to the lower effective recombination rate in the selfing species compared to the outcrossing species.

Note that I don't doubt at all that the data are robust to these questions with regard to the choice of $h=f(s)$ over any constant h model; the issue is the robustness of the specific details of $h=f(s)$.

One of the motivations for the research is to test existing evolutionary models of dominance. I'd say that Fisher's model has been dead for about 50 years, and certainly since Orr 1991, so there's no need to spend a lot of time on it. The authors argue that the presence of a relationship between h and s across gene categories implies that the metabolic theory can't be the whole story. They extend an existing model to incorporate variation among genes in

optimal expression levels, showing that it predicts the relationships they observe. This is an interesting and valuable addition to the dominance theory literature.

We thank reviewer 3 for his/her thoughtful and constructive comments. We have revised the manuscript in light of these comments and believe that our new analyses and results have strengthened our manuscript.

Minor things:

I found the reference to the omnigenic model at line 287 to be mysterious; I don't see how the authors's results relate to that model.

We removed the reference to the omnigenic model.

Line 475: when two species do not share ancestral polymorphism.... Add here that *A. lyrata* and *A. thaliana* fit this description. Most readers will not already know this.

We added the following sentence (line 527-530):

The species pair *A. thaliana* and *A. lyrata* meets this assumption, since the probability for shared ancestral polymorphisms is negligible small and allele frequencies are highly uncorrelated (Novikova et al. 2016). The log-likelihood of the full model can thus be summed according to equation (3).

Fig 1 A and B: I spent quite a while confused by these figures until I realized that each set of bars represents a totally different unspecified DFE. Maybe there's a way to say that in the in-figure key.

In the cartoon example of Fig 1 A and B, the DFE consists of a single value of s , i.e. every mutation has the same value of s . We added this value of s to the in-figure key.

REVIEWERS' COMMENTS:

Reviewer #1 (Remarks to the Author):

I think that this manuscript is ready to be published.

Reviewer #2 (Remarks to the Author):

I appreciate the thoughtful responses the authors gave to all the reviewer comments, and have no further criticisms.

Reviewer #3 (Remarks to the Author):

The revised manuscript is terrific. I really appreciate the authors' efforts to address the issues raised in the reviews, particularly variation in $h|s$ and alternative forms of $h=f(s)$. New figure 3 greatly improves the manuscript. My only remaining suggestion is to add a sentence justifying the relationship in eq (1). Something like "We chose to model the relationship between dominance and selection coefficient in this manner because"

There are a few details of the figures that need attention:

Figures 2b-d: There are some missing gridlines— looks like something went slightly wrong in Illustrator.

2d: legend says tan, figure shows blue.

Figure 3b: The plotted points don't seem to be points.

Fig 3b: "True" mean $h-s$ relationship is confusing to me; maybe this could be "Simulated mean $h-s$ relationship."

Fig 4a: The blue lines are missing?

RESPONSE TO REVIEWERS' COMMENTS:

Reviewer #1 (Remarks to the Author):

I think that this manuscript is ready to be published.

We thank the reviewer for the positive comment.

Reviewer #2 (Remarks to the Author):

I appreciate the thoughtful responses the authors gave to all the reviewer comments, and have no further criticisms.

We thank the reviewer for the positive comments and appreciating our response to his/her previous comments.

Reviewer #3 (Remarks to the Author):

The revised manuscript is terrific. I really appreciate the authors' efforts to address the issues raised in the reviews, particularly variation in $h|s$ and alternative forms of $h=f(s)$. New figure 3 greatly improves the manuscript. My only remaining suggestion is to add a sentence justifying the relationship in eq (1). Something like "We chose to model the relationship between dominance and selection coefficient in this manner because"

We thank the reviewer for the positive comments and appreciating the extra analyses that we did for the revised manuscript. We agree that an extra sentence justifying the relationship in equation 1 would be helpful. We now add the following sentence on lines 128-130:

"We chose to model the h - s relationship in this manner to allow for more deleterious mutations to be more recessive than less deleterious mutations, as is suggested by experimental data¹¹⁻¹³."

There are a few details of the figures that need attention:

Figures 2b-d: There are some missing gridlines— looks like something went slightly wrong in Illustrator.

We do not see any missing gridlines when inspecting the file on our computers. We will pay close attention to this issue when checking the final proofs.

2d: legend says tan, figure shows blue.

We fixed this error.

Figure 3b: The plotted points don't seem to be points.

Again, when we open the file on our computers we do not observe this issue. We will check the proofs for this problem.

Fig 3b: “True” mean h-s relationship is confusing to me; maybe this could be “Simulated mean h-s relationship.”

We changed the wording to “Simulated mean h-s relationship”.

Fig 4a: The blue lines are missing?

The lines are not missing but partly overlap with the grey lines of the genome-wide estimates and are thus hard to see. We added a clarification in the figure caption of Fig.4a, lines 887-888:

“The blue lines in the middle and right panel strongly overlap with the grey lines of the genome-wide estimates.”